# Spatial Metagenomic Analysis in Understanding the Microbial Diversity of Thar Desert

**DOI:** 10.3390/biology11030461

**Published:** 2022-03-17

**Authors:** Jagdish Parihar, Suraj P. Parihar, Prashanth Suravajhala, Ashima Bagaria

**Affiliations:** 1Department of Physics, Manipal University Jaipur, Jaipur 303007, India; jagdish.parihar@muj.manipal.edu; 2Wellcome Centre for Infectious Diseases Research in Africa (CIDRI-Africa), Institute of Infectious Diseases and Molecular Medicine (IDM), Division of Medical Microbiology, Faculty of Health Sciences, University of Cape Town, Private Bag X3, Rondebosch, Cape Town 7701, South Africa; suraj.parihar@uct.ac.za; 3Bioclues.org, Vivekananda Nagar, Kukatpally, Hyderabad 500072, India; 4Amrita School of Biotechnology, Amrita Vishwavidyapeetham, Amritapuri Campus, Clappana P.O., Kollam 690525, India

**Keywords:** metagenomics, extremophiles, microbial diversity, soil bacteria, evolutionary taxonomy

## Abstract

**Simple Summary:**

We present a systematic investigation of the distribution of microbial communities in arid and semi-arid regions of Thar Desert Rajasthan, India. Their responses in multiple environmental stresses, including surface soil, surface water and underground water were evaluated. We further assess the biotechnological potential of native microorganisms and discover functional species with results providing a detailed understanding of the abundance of microbial communities in these regions, associated with various stress-related biogeochemical and biotechnological processes. We hope our work will facilitate the development of effective future strategies for the use of extremophiles in complex environments.

**Abstract:**

The arid and semi-arid regions of Rajasthan are one of the most extreme biomes of India, possessing diverse microbial communities that exhibit immense biotechnological potential for industries. Herein, we sampled study sites from arid and semi-arid regions of Thar Desert, Rajasthan, India and subjected them to chemical, physical and metagenomics analysis. The microbial diversity was studied using V3–V4 amplicon sequencing of 16S rRNA gene by Illumina MiSeq. Our metagenomic analyses revealed that the sampled sites consist mainly of Proteobacteria (19–31%) followed by unclassified bacteria (5–21%), Actinobacteria (3–25%), Planctomycetes (5–13%), Chloroflexi (2–14%), Bacteroidetes (3–12%), Firmicutes (3–7%), Acidobacteria (1–4%) and Patescibacteria (1–4%). We have found Proteobacteria in abundance which is associated with a range of activities involved in biogeochemical cycles such as carbon, nitrogen, and sulphur. Our study is perhaps the first of its kind to explore soil bacteria from arid and semi-arid regions of Rajasthan, India. We believe that the new microbial candidates found can be further explored for various industrial and biotechnological applications.

## 1. Introduction

Extreme environments refer to the habitats that are detrimental to living organisms. These habitats include various physicochemical parameters such as temperature (>40 °C & <5 °C), pH (acidic pH < 5, alkaline pH 9), nutrient content, pressure (0.1 MPa), salinity (35%) and radiation (aw 0.8) [1]. The organisms flourishing under these extremities are called extremophiles. The great Indian Thar desert spans over the vast arid and semi-arid regions in the north-western part of India and offers temperature fluctuations varying from 5 to 55 °C. This extreme habitat poses a threat to most life forms but is also a haven for the flourishing diversity of thermophiles [2]. The optimal temperature range further breaks down the classification of these surviving microbes as mesophiles (37–40 °C); moderately thermophiles (40–60 °C), thermophiles (60–80 °C) and hyperthermophiles for microbes sustaining above 80 °C [3,4]. Thermophilic bacteria thrive in habitats such as hot springs, geothermal vents and desert soil [5,6]. In terms of soil microbial community structure and function, it has been shown that desert biomes vary significantly from other biomes [7]. The most recent comparison of soil microbial population across different biomes using metagenomic sequencing depicts the lower values of phylogenetic and functional diversity crossing several main taxa [8]. Microbiologists have been studying the diversity of biomes across plants and animals for a long time and it is only in the recent past that the scientific community has started exploring soil microbiomes [9,10]. Microorganisms perform vital functions in environmental ecosystems and are sensitive to environmental stress [11,12]. Microbial diversity of soil and other extreme environments, however, is still limited to only 1–3% of cultivable soil microbes and is estimated to comprise approximately 4000–5000 different bacterial genomic units in 1 gm of soil [13,14]. The recent advancement in next generation sequencing (NGS) has heralded rigorous microbial studies, thereby providing new insights towards understanding microbial diversity, stability, and adaptability under extreme environmentally stressed conditions [15,16,17]. Microbes within the natural surroundings exist in complicated communities marked by intricate webs of different interactions [18]. The ecological interactions among microbial communities possess properties of positive cooperation or negative antagonism that have an impact on overall microorganism composition [19]. With soil samples possessing huge diversity across different environmental conditions [20,21], it is estimated that there would be approximately 3500–8800 genome equivalents in a pasture soil sample. Given the relative abundance of about 10,000 different species in the soil ecosystem, this immense diversity is a challenge for researchers trying to categorize microbial communities into distinct species [22]. Minor fluctuations in the soil environment tend to have a major impact on the diversity and community structure of the species [23]. Heavy metals, organic contaminants and pesticides are just a few environmental variables which affect the structure of the microbial community, with the parameters that influence microbial composition include particle size, permeability, porosity, water content, mineral composition, and plant cover.

The 16S and 18S rRNA gene sequencing gives insights into the bacterial and fungal diversity, respectively, whereas sequencing the inter transcribed sequence (ITS) region gives information about eukaryotic diversity [24]. With 16S rRNA gene sequencing being a commonly used tool for identifying bacteria, tracing phylogenetic relationships between bacteria, and identifying bacteria from various sources such as soil and water play an important role [25]. On the other hand, the 18S rRNA gene is a prevalent molecular marker for biodiversity research due to its highly conserved nature and slow evolution and thus assists in the analysis of species [26].

The bacterial 16S rRNA gene has nine hypervariable regions (V1–V9). We have selected the V3–V4 region because of the heterogeneity of the region and low evolution compared to other hypervariable regions, which can help better identify the organism. The sequences obtained from the Greengenes, NCBI, RDP, and SILVA databases representing the primers chosen for V1–V2 were significantly smaller (30.3%) than that of V3–V4 (90%), V4 (90.9%), and V4–V5 (87.8%) [27,28]. Whereas V3–V4 fragments do not allow distinguishing some species-level operational taxonomic units (OTU) merging into single species, the clusters could be determined at the level of genetic distances of V3–V4 fragments. The main reason why databases have a major proportion of 16S sequences submitted from the V3–V4 region is because they are produced by V3–V4 region primers (most used primers in microbial community analysis).

Though thermophiles have been widely studied across the globe, an understanding of the biodiversity in desert biomes, specifically with respect to NGS, is scarce. The majority of protocols used for diversification are limited to morphological features. The advancement in next generation sequencing techniques have enabled researchers to characterize the microbial population utilizing these metagenomic techniques for both cultivable microorganisms as well as those that cannot be cultured. These culture independent techniques based on DNA sequencing, do not depend on morphological parameters, and have filled the research gap to a large extent, but they have their own limitations. Asian countries harbour many hot springs which are good sources of thermophiles [29] and researchers have attempted to unlock the microbiome of various soil samples collected from different sites based on a non-redundant inter-transcribed sequence (ITS) region [30]. For example, the impact of agricultural management on soil microbial communities by generating amplicon and shotgun metagenome libraries using metagenomic approaches were studied [31,32]. The first culture independent study of microbial diversity within the hot spring hypersaline lakes Magade and little Magadi in Kenya was carried out and their metagenomic study mainly focused on the comparison of total versus active microbial communities [33]. Another study showed the identification of the cellulolytic systems and addressed the strategies adopted for the discovery of new cellulases in the field of metagenomics, emphasizing on thermophilic microorganisms [3,34]. Furthermore, comparative metagenomic profiles not only help understand mechanisms for stress but also microbial adaptation to such impending stress associated with ecological patterns such as extreme hot/cold deserts and other niche communities [5]. Nevertheless, very small efforts have facilitated understanding microbes from varied samples in Indian conditions. For example, a study with soil and water samples could allow us to identify the genetic determinants between metagenomes of desert and non-desert soils. In bridging such gaps, our current metagenomic approach focuses on microbial diversity and their abundance in arid and semi-arid regions of the great Indian Thar Desert, Rajasthan, India, by inferring amplicon-based strategy. The goal of this study was to find the abundance of taxa present in the samples targeting the V3–V4 specific gene amplicon sequencing of 16S rRNA across six vivid samples (5 soil and 1 water sample). To the best of our knowledge, this study is perhaps the first of its kind, where thermophilic samples from extreme arid and semi-arid regions of Rajasthan, India have been analysed. We present the unique bacterial taxonomic clades obtained from our samples and discuss the challenges and implications of bacteria living under these extreme conditions.

## 2. Materials and Methods

### 2.1. Site Description and Sampling

The soil and water samples were collected maintaining heterogeneity from arid and semi-arid zones of the Thar during peak summer (June) from Jaisalmer, Barmer, Pachpadra and Pokhran, districts of Rajasthan, India (Appendix A). The soil samples were collected by sterile spatula and bottle (1 kg for each soil sample and 1 litre water sample), marking the area to 1 × 1 × 1 foot (length, breadth and height) to remove the uppermost soil which may be influenced by environmental factors. The environmental parameters such as elevation, humidity, precipitation and temperature in the summer season were also measured during sampling. This belt is extremely dry and the temperature ranges from 40 °C to 60 °C. At the time of sampling, the average temperature of the soil sample measured was 55 °C. For temperature as an important factor, we observed temperature for three days and collected the samples at the highest temperature. There is no plant coverage in this study as all the samples were collected from an area that is barren or has low human and animal interference. A total of 5 soil samples were covered under the study and 1 water sample as negative control (water samples were collected from 6 sites from natural water reservoirs. At every single site 6 samples were collected and a total of 36 samples from 6 different sites were collected and pooled). In the case of a water sample (1 L each from 6 different sites was collected and pooled). The soil samples were collected and stored at 4 °C. These samples were transferred to the laboratory for further processing.

### 2.2. Physicochemical Analyses

The collected samples were filtered through 2 mm mesh and air dried to identify physicochemical properties. The chemical characteristics such as Total Organic Carbon (TOC) was determined by a rapid titration method. Available phosphorus and potassium were measured by spectrophotometer and flame photometer and micronutrients content (iron, copper, zinc and manganese) were detected through atomic absorption spectroscopy (Agilent Technologies, 200 Series AA, Santa Clara, CA, USA) following content using standard operative methods. The physical characteristics of the samples such as pH and EC were evaluated by a multi parameter tester [35,36,37,38]. For all components analysed in this research, analytical uncertainties are <5 percent (Appendix A).

### 2.3. DNA Extraction, PCR Amplification, and Illumina MiSeq Sequencing

The Fast DNA Spin Kit for Soil (Qiagen, La Jolla, CA, USA) was used to extract the genomic DNA from all field samples and incubated samples according to manufacturer instructions. A 25 ng of DNA was used to amplify hypervariable V3–V4 regions of total 16S rRNA genes from sediment and aqueous samples [39]. The reaction includes KAPA HiFi HotStart Ready Mix and 100 nm final concentrations of modified 341F and 785R primers [40]. The flowchart of the sequencing protocol followed is shown in Figure 1. The primers were designed to identify areas with amplicon length compatible with the current Illumina MiSeq sequencing platform (2 × 300 bp) by first examining the 16S rRNA gene in silico, and then targeting conserved and variable regions. The conserved regions were screened against the non-redundant database of bacterial model systems (*E. coli*) of NCBI BLAST and validated for their suitability in amplifying the 16S rRNA gene V3–V4 region. The full-length primers were designed using dual-index methods [41]. While designing full-length primers, each forward and reverse primers have an adaptor sequence (provided by NGS Company), four random nucleotide sequences followed by an eight-nucleotide indexing sequence, followed by a 16S rRNA V3–V4 unique forward priming sequence of 17 nucleotides (CCTACGGGNGGCWGCAG) and the reverse primer sequence containing 21 nucleotides (GACTACHVGGGTATCTAATCC). The purpose of the adapter sequence is to allow the DNA sequence attachment to the MiSeq flow cell and the connector sequence to increase the overall temperature of melting. These degenerate primers collections were further used with site-specific adaptor sequences to demonstrate the functional heterogeneity of soil and water samples. PCR was subsequently set with an initial denaturation of 95 °C for 5 min followed by 25 cycles of 95 °C for 30 s, 55 °C for 45 s and 72 °C for 30 s and a final extension at 72 °C for 7 min. The amplicons were purified using Ampure beads to remove unused primers. An additional 8 cycles of PCR were performed using Illumina barcoded adapters to prepare the sequencing libraries. The libraries were quantitated (Appendix A), using Qubit DNA HS quantitation assay (Thermo Scientific, Waltham, MA, USA), which specifically quantitates dsDNA assay.

### 2.4. Technical Validation

The filtered dataset for the V3–V4 16S rRNA fragment included 1,362,076 reads with the average length of 301 base pairs. We obtained 6 clusters with a total of 617 species-level taxonomic clades wherein the V3–V4 dataset included 681,038 reads with an average length of 344 base pairs. For analysing the convergence, proportions of underestimated OTUs were checked for biological studies and validation. We removed all the duplicate, unwanted and unknown sequences and analysed 656 different sequences. Out of these, our analyses confirmed 64 unique sequences. Alpha diversity richness was demonstrated by heat maps and principal component analysis (PCA) plots while beta diversity richness was indicated by a phylogenetic tree.

The two sets of reads for every pattern (forward and reverse reads) were blended using (make.contigs) command. Sequences that failed to fulfil any one of the following standards were excluded, whereas an average length of 344 bases was analysed for the presence of ambiguities and nucleotide mismatch (>1) to the primer using (screen.seqs) command and duplicate sequences were further removed using (unique.seqs) command from the analyses. The pcr.seqs was used to infer V3–V4 vicinity of 16S rRNA genes wherein unique sequences were aligned against the SILVA reference database. By using (filter.seqs) command, gap characters had been pulled out without losing any facts in alignment of columns. We then used the (pre.cluster) command to allow at most two variations between sequences followed by identifying chimeras using UCHIME algorithm (chimera.uchime). The sequences had been taxonomically classified using the Naive Bayesian classification with 80% confidence threshold via the use of (classify.seqs) command. The sequences that were not categorized to any one of the domains (unknown) or labelled in Chloroplast, Mitochondria, Eukaryota, and Archaea were eradicated using (remove.lineage) command, wherein each of these sequences are counted as a separate community.

### 2.5. Phylotype Analysis

The analysis of sequences into phylotypes as per taxonomic characterization was carried out using distance matrix sequences clustered into OTUs with cut-off 97% similarity (3% dissimilarity) using (dist.seqs) and (cluster) commands. The cut-off numbering of the phylotypes rises to four, which compared to the arranged level utilized in (cluster.split) command. In each OTU, the wide range of sequences was identified by using (make.shared) command and the unique OTUs were arranged using the (classify.otu) command.

### 2.6. Sequence Data Quality Control

The sequence data quality was checked using FastQC [42] and a consensus was reached using MultiQC [43]. The data was checked for base call quality distribution, bases% above Q20, Q30, GC%, and sequencing adapter contamination (Appendix A). All the samples have passed QC threshold (Q20 > 95%) with the average GC percentage found to be 55% among all the samples.

### 2.7. NGS Analyses

The reads were trimmed (20 bp) from 5′ end to remove the degenerate primers. These trimmed reads were then processed to remove adapter sequences and low-quality bases using Trim Galore [44]. The quality of raw reads was ensured using FastQC [42] and the analysis was carried out using Mothur pipeline [45]. The pairs were aligned with the contigs screened for errors and only those between 300 bp and 532 bp were retained. While contigs with ambiguous base calls were rejected, the high quality contigs were retained for checking identical sequences, and duplicates were merged. The Contig length distribution was shown in (Appendix A) Although the primers for the experiment were designed for 16S bacterial rRNA, the chances for non-specific amplification of other regions cannot be neglected. To correct this, we aligned the contigs to a SILVA rRNA database for 16S rRNA. Depending on the variable region being amplified, most and contigs aligning to other regions on the database were discarded. Further the gaps and the overhangs from the contigs were removed and processed for removal of the artifacts or chimeras, which may have been formed due to PCR errors. UCHIME algorithm [46] was used to flag contigs with chimeric regions. A known reference of all the chimeric sequences was used to identify and remove possible chimeric sequences and shown in Figure 1.

The filtered contigs were processed and classified into taxonomic outlines based on the Silva v.132 database [47]. The contigs were then clustered into OTUs and the abundance was further estimated. Sample dissimilarities were calculated by Bray–Curtis matrices. The Mothur was used to generate curves of rarefaction (Appendix A) and Alpha-diversity [48,49]. With non-metric multidimensional scaling (NMDS) methodology [PAST software program, (v.3.16)] employed to display bacterial communities. The heatmaps were generated for top 20 phylum and genus normalized OTUs using heatmap (R package), while principal component analysis (PCA) was carried out using ClustVis (Beta) [50].

## 3. Results and Discussions

### 3.1. Microbial Structure, Diversity and Richness

Microbial community structures from different environmental samples were diverse, with relatively lower richness and diversity found in soil samples as compared to water samples. *Proteobacteria* were widespread and dominated microbial communities in all samples. The structure of microbial communities changed considerably during cultivation. In our soil and water samples, we found a microbiome consisting mainly of Proteobacteria (19–31%) in abundance, followed by unclassified bacteria (5–21%), then Actinobacteria (3–25%), Planctomycetes (5–13%), Chloroflexi (2–14%), Bacteroidetes (3–12%), Firmicutes (3–7%), Acidobacteria (1–4%) and Patescibacteria (1–4%), respectively, shown in (Appendix A). The heatmap shows distribution of top 20 Phyla with proteobacteria being abundant in Sample-6 and Actinobacteria abundant in Sample-2, compared to other samples shown in Figure 2a and Figure 3a. Predominantly, we found Proteobacteria and Acidobacteria which have potential biotechnological applications such as paper and pulp industry, textiles industry, soap and detergents industry, leather processing industry, dairy industry, food and beverages, dairy industry, baking industry, meat industry, pharmacology and drug manufacturers, biofuel production, biomining, medicines, bioremediation, anti-inflammatory, antimicrobial, anti-cancer and also exploitation for different therapeutic applications.

The heatmap represents the distribution of the top 20 Genus, Gemmataceae; uncultured is more abundant in Sample-1. Similarly, Curvibacter is more abundant in sample-6 as compared to other samples shown in Figure 2b and Figure 3b. In the heatmap, blue colour signifies less abundant, light blue and pale-yellow mean moderate and red colour signifies the most abundant of the species. This study was carried out to determine the overall similarity between the composition of microbial population in soil and water samples.

The stacked column bar graph represents the microbial community composition at the phylum level across all samples used in this study. Taxonomic identities that could not be shown to the respective level of resolution were considered as “unclassified” within their corresponding domain. Relative abundance data were analysed using MG-RAST against the SEED database and then visualized using Microsoft Excel. Sample names are included in the plot (sample-1 = soil; sample-2 = soil; sample-3 = soil; sample-4 = soil; sample-5 = soil; sample-6 = water.

Predicted community composition of each assembly from sample-1 to sample-6, coloured by genus, with proportion defined by summed length of contigs assigned to that genus. Deeper subsampling dramatically changes the community composition.

Microbial population data consisted of (percent) normalized OTUs. Graphical representations along the PC1 (33.8%) and PC2 (26.3%) axis had a 60.1 percent average variance. Sample 6 was widely distributed in different taxa as compared to sample 3, followed by sample 4, 5, 2 and 1, with sample 1 being the least abundant and shown in Figure 4. The Venn diagram showing the number of taxa shared by, or unique to, the different samples were built using the Venn and galaxy project (URL: usegalaxy.org, accessed on 27 February 2022) [51,52]. The most common species in all samples was *Wigglesworthia brevipalpis,* shown in (Appendix A).

### 3.2. Alpha Diversity and Beta Diversity Analysis

Alpha diversity is a measurement of richness and relative abundance of bacteria within the sample. Chao1 and ACE indices represent the richness of the samples and Shannon, Simpson, InvSimpson and Fisher indices represent both richness and relative abundance shown in Figure 5a. The alpha diversity calculations were performed using Phyloseq an R package (vegan 2.4.4) for metagenome analysis. Using the alpha rarefaction.py script in QIIME, we calculated alpha OTU diversity by randomly sub-sampling (without substitute) each soil and water sample. In addition to the observed amount of OTUs, the Chao1, Shannon, phylogenetic diversity (PD) and Evenness indices of Faith were calculated.

The rarefaction gives the diversity but not the rich coverage and, hence, the differences in coverage are expected after sampling a large number of individuals with select 1000 sequences from each sample, and so the average was calculated. While the quantitative estimation of coverage served as a basis for the adjustment of tests from the distance matrix/phylip tree, we ascertained the measures from binned fragments.

### 3.3. Comparative Analysis from Different Deserts

In the recent past, the investigation of soil microbiomes in different geographical regions of the world has been carried out by NGS. Some of the studies mentioned here show the presence of a diverse microbial population in desert soils. For example, the arid desert soils of Kazakhstan, Atacama Desert, Namib, Gobi, Thar, Negev, Sonoran, and Mojave deserts are highly diverse, dry and arid with low humidity and intense radiation, while these physical parameters allow special microbial communities to flourish in these extreme conditions. The dominant phyla present in these deserts are mainly Proteobacteria, Actinobacteria, Firmicutes, Chloroflexi, Verrucomicrobia, Acidobacteria, Planctomycetes, cyanobacteria and Bacteroidetes, respectively. The functional genes related to these taxa play an important role in biogeochemical cycles [53,54,55,56,57,58,59,60]. In our study we have also shown that Proteobacteria and Actinobacteria phyla are most abundant beside other phyla, respectively, among different soil samples. Therefore, we can conclude from above studies that our microbial diversity in the Thar desert is similar to the diversity of deserts in different geographical regions, though a thorough study is required to establish certain facts and figures with other desert biomes.

**Figure 5 biology-11-00461-f005:**
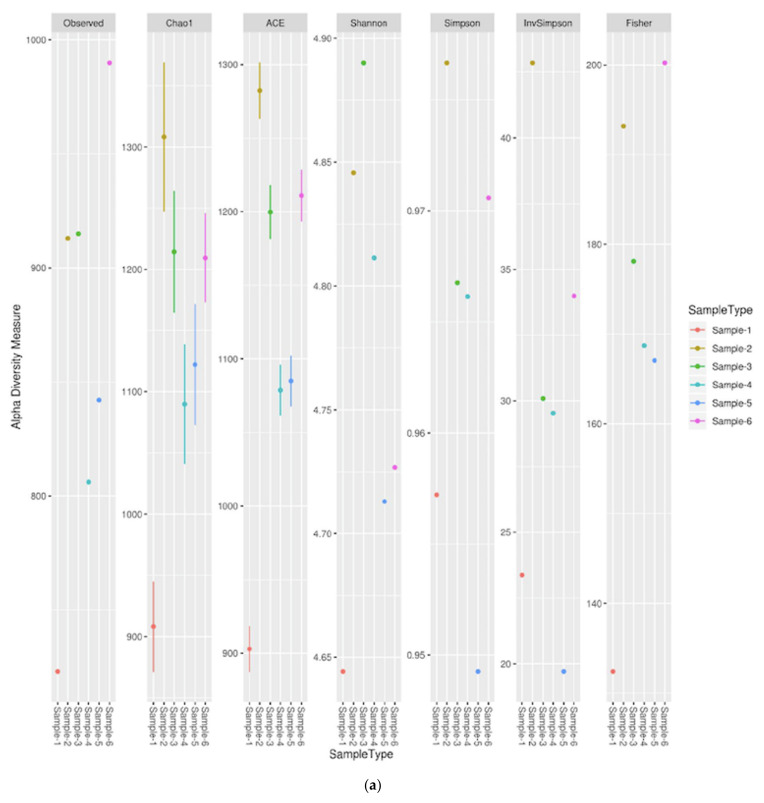
(**a**) Alpha diversity measurements relative to various statistical thresholds. Beta diversity refers to the ratio between local or alpha diversity and regional diversity. This is the diversity of species between two habitats or regions. The relationship between species and genomes is identified by the phylogenetic tree of life. The centre displays the last universal ancestor of all life on earth. The three domains of life are shown by different colours: pink represents eukaryota (animals, plants and fungi); blue represents bacteria; and green represents archaea. Taxonomic clades of six samples spanning from arid and semi-arid habitats showing the root clade. These figures are produced by iTOL and shown in Figure 5b [61]. (**b**) Taxonomic clades of six samples spanning from arid and semi-arid habitats showing the root clade [Color code: Bacteria-purple; Pink-Eukaryotes; Green-Archaea].

## 4. Conclusions and Future Aspects

The present study thus presents a systematic investigation of the distribution of microbial communities in arid and semi-arid regions of Thar Desert Rajasthan and their responses in multiple environmental stresses, including surface soil, surface water and underground water. It also deals with high temperature, salt concentration, and pH stress to assess the biotechnological potential of native microorganisms and to discover functional species. Our results provide a detailed understanding of the abundance of microbial communities in these regions, associated with various stress-related biogeochemical and biotechnological processes that will facilitate the development of effective future strategies for the use of extremophiles in complex environments.

The results obtained are among the few metagenomic diversity studies available for this region. The study displays the diversity of largely unknown soil microbiomes by preliminary analysis and further exploration is required to quantify the presence of microbes and their mechanism of function and stability. In this pilot study, we have majorly focused on how different environmental conditions affect the microbiome of any region. We have covered the maximum area for the study by collecting samples from different geographical locations which consist of different microbial populations. Hence, we have shown the microbial composition of various sites. The characteristics of soils under various lands are usually different. The proportion of habitat disturbance produced by variations in land-use management methods, which impacted soil characteristics, was found to influence microbial diversity. This research offers up new possibilities for metagenomics research in Rajasthan’s semi-arid habitats, and it contributes to the discovery of soil bacteria that are important for ecosystem processes.

Further evaluation of our samples will help us to enhance our knowledge of microbial composition and features in soil and learn about their one-of-a-kind pathways and metabolism. Future efforts will be directed towards identifying species and metabolisms associated with it in each geographic region. In addition, the datasets can be used in future for worldwide soil metagenomic projects for comparative purposes. Additional experimental and sequencing efforts will help in understanding soil microbial dynamics and the specific interacting microbes that would lead to enhancement in cutting-edge agricultural and soil sustainability practices.

## Figures and Tables

**Figure 1 biology-11-00461-f001:**
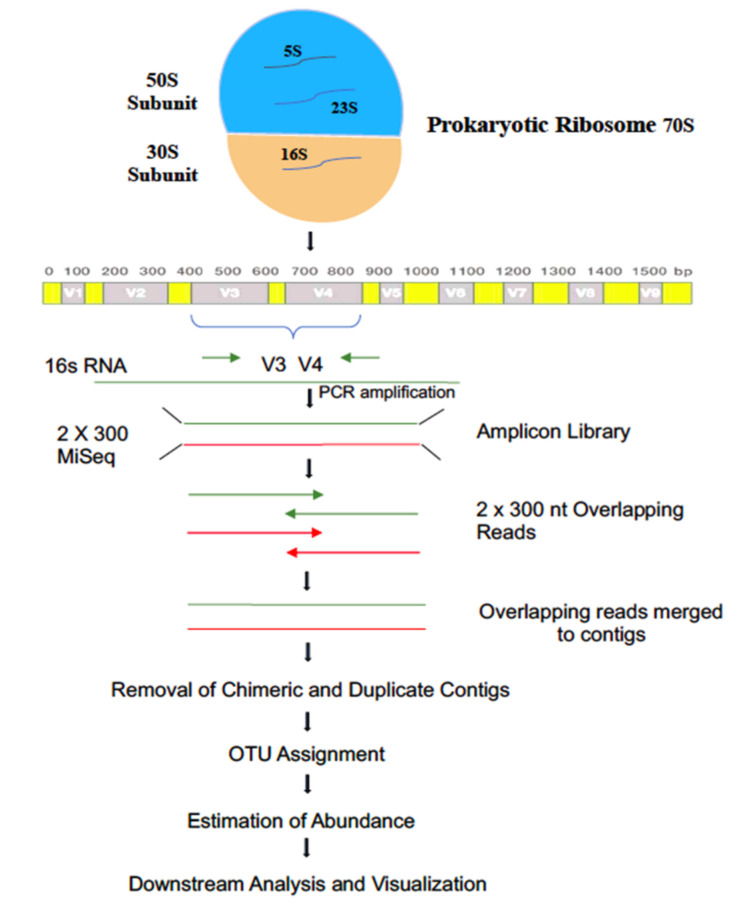
Flowchart depicting 16S rRNA metagenomic workflow.

**Figure 2 biology-11-00461-f002:**
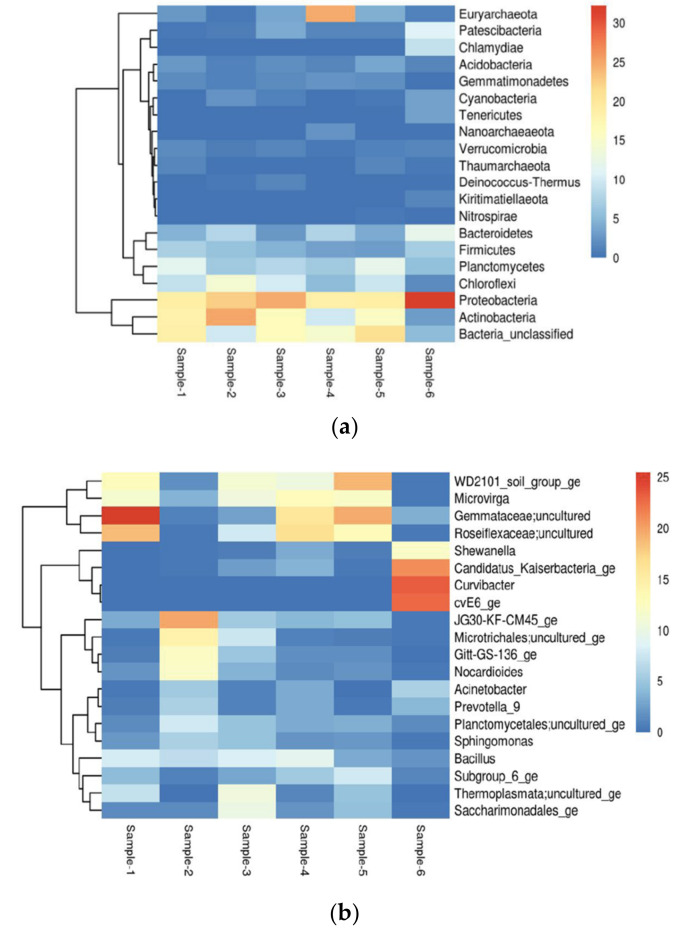
(**a**) Top 20 Phylum abundance distribution. (**b**) Heatmap representing distribution of top 20 Genus.

**Figure 3 biology-11-00461-f003:**
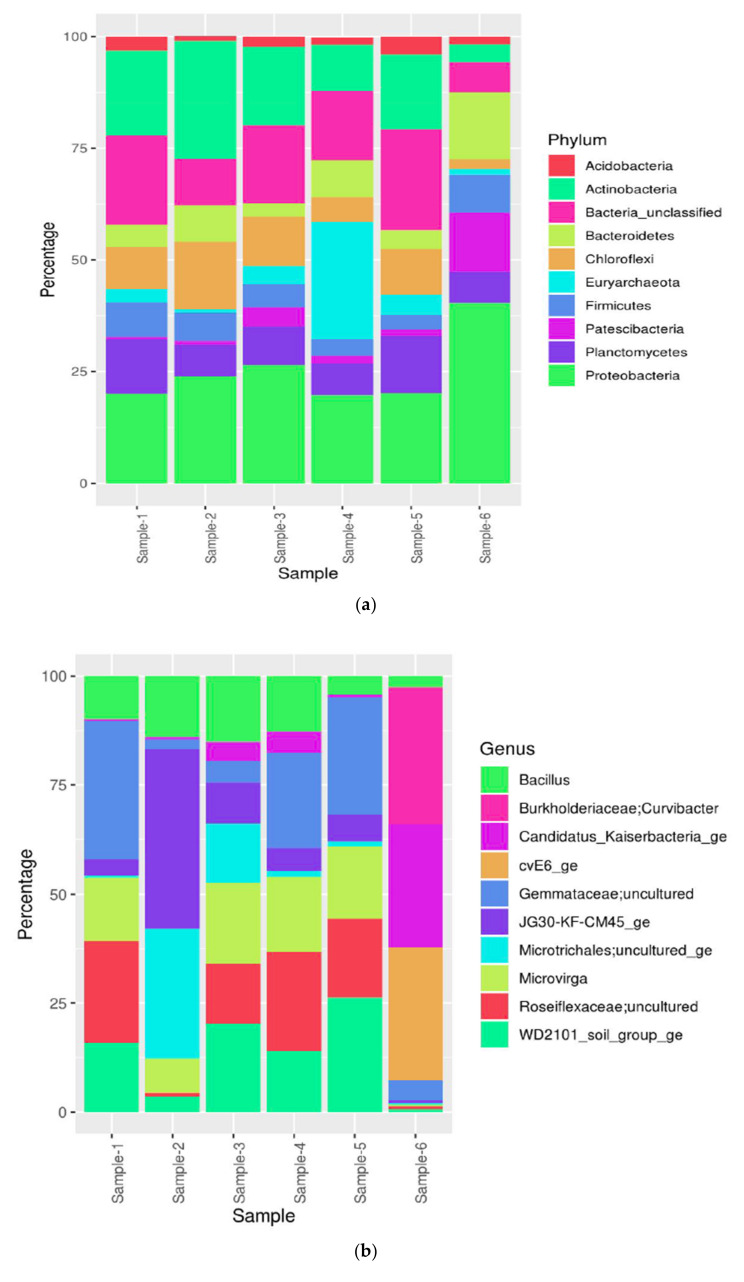
(**a**) Top 10 Phyla abundance distribution. (**b**) Top 10 Genus abundance in coloured maps.

**Figure 4 biology-11-00461-f004:**
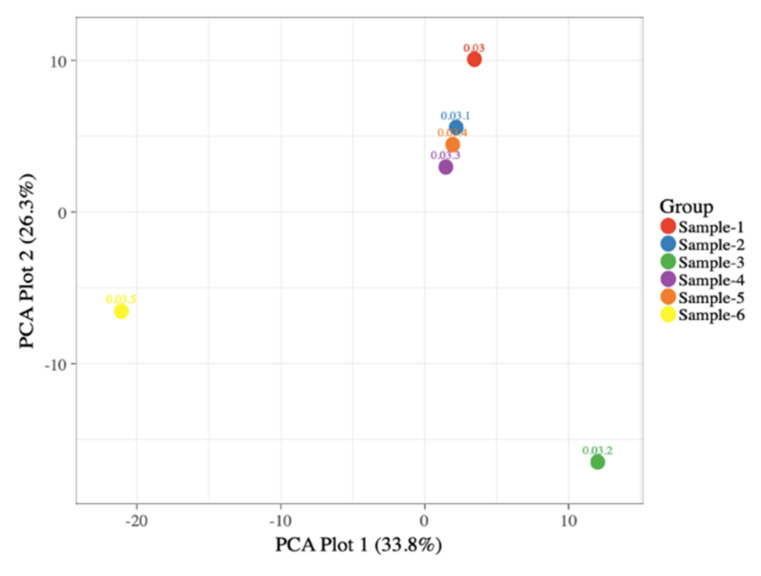
Principal component analysis (PCA). PCA plot illustrating correlation among six samples (five soil samples and one water soil sample) from arid and semi-arid regions of Rajasthan.

## Data Availability

The nucleotide paired-end sequences have been deposited in NCBI SRA database with accession ids: SRR14334654, SRR14334655, SRR14334656, SRR14334657, SRR14334658, and SRR14334659 for 6 samples for 16S rRNA (V3-V4 region) amplicon reads under the Bio Project PRJNA725519.

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
