# Peer review of "Spatial Metagenomic Analysis in Understanding the Microbial Diversity of Thar Desert"

_biology, 2022, doi:10.3390/biology11030461_

Round 1
Reviewer 1 Report
The manuscript entitled” Spatial Metagenomic Analysis in Understanding the Microbial Diversity of Thar Deserts” conducted the metagenomic analysis to identify microbial diversity in arid and semi-arid areas of Rajasthan, India, where they found proteobacteria is most abundant. While I appreciate the amount of this work, overall, I do find the entire research ill-designed and performed, provides minimal insights into this field. No critical conclusions and findings were reported. Thus, the current manuscript is not suitable for publication.
Major points:
1 The sampling process and design are very inadequately executed. Based on its supplementary information, no details of site locations and environmental characterization is provided here. It only generates latitude and altitude. What about elevation? What about soil relative humidity and its average? Is there plant coverage? What about the local climate influence? Even for temperature, have the authors measured the soil temperature for a certain period of time instead of one point during sample harvest? Even the characterization of aridity itself lacks important details. It is not hard to deduce that in arid ecosystems, regions of higher aridity correlate with decreased microbial taxonomic enrichment and apparent changes regarding phylogenetic composition. Why not consider these important attributes in more detail?
In addition to this, they simply claimed the number 6 sample is water control. Why use water? What kind of water? Why not set up water control for each sampling location? Control, in this case, matters given the authors only listed 6 samples for further analysis, whereas many published findings used dozens of samples managing to maintain the diversity (see examples in PMID: 28593197, 31608023), not to mention there are no biological replicates (triplicates at least) sampled for microbial composition analysis.
2 Based on the authors’ analysis in Figure 2 and 3, I failed to see any significant changes in terms of both phylum and genus abundance distribution among the 5 soil samples. Even when compared with the water sample which should theoretically be distinct, there are no major differences or pattern changes in my point of view. Even the PCA plot in Figure 4 seems to recognize some differences in sample 6, what about sample 3? How to explain the discordance among the 4 samples all collected from semi-arid areas? And how come the difference only appear on PCA plot but failed to emerge from other analysis?
3 All the figures presented are of extremely low quality. The supplemental data does not contain useful supporting information, not to mention the conclusion (lane 323-325) was drawn concerning the microbial diversity similar to that of deserts in different geographical regions, rendering the entire study worthless and fundamentally insignificant.
Reviewer 2 Report
This is a standard Metagenomic study. The experimental design and data analysis was straightforward, and the data from desert soil sample is of general interest to biologists and ecologists. Two minor points: 1) to better understand the data, the authors should provide more details about the soil samples (depth, surround environments, collecting size etc) in the manuscript; 2) the introduction section is a bit too long compared to the short discussion section.
Reviewer 3 Report
The manuscript entitled "Spatial Metagenomic Analysis in Understanding the Microbial Diversity of Thar Desert" is well written and the data presented is scientifically sound and relevant. A few details can be improved, such as the quality of some figures and a few spelling checks. Also, supplementary tables 3-6 are missing, couldn’t check those data. The word file “Supplementary Table 1-6.docx” had only supplementary tables S1 and S2. In addition, supplementary figures are unreadable, except Figure S2. It would be nice to have a map of the region, maybe around Page 3, line 113, showing the sampling sites and a brief description of each one. Photos of the sampling sites would be a wonderful addition to the main text or in the supplementary material.
The minor comments below reflect mostly some text editing issues and a few details to be corrected/improved:
1) Page 1, line 29: Add a period after the first sentence.
2) Page 1, line 35: Maybe replace “normal life” by most life forms
3) Page 1, line 38: There is an extra space before the word “thermophiles”
4) Page 2, line 49: There is an extra space before the word “approximately”
5) Page 2, line 68: There is an extra space before the word “and”
6) Page 2, line 79: There is an extra space before the word “sequences”
7) Page 3, line 111: There is an extra space before the word “metagenomes”
8) Page 3, line 112: There is an extra space before the word “focuses”
9) Page 3, line 116: There is an extra space before the word “and”
10) Page 3, line 146: There is an extra space before the word “The”
11) Page 5, line 185: There is an extra space before the word “against”
12) Page 5, line 188: There is an extra space before the word “the”
13) Page 5, lines 180-203: There are multiple commands listed in sections “2.4 Technical validation” and “2.6. Sequence Data Quality Control”. Do they relate to the software “R”? Please describe.
14) Page 5, line 214: There is an extra space before the word “The”
15) Page 6, line 228: There is an extra space before the word “OTUs”
16) Page 6, line 243: Replace the word “than” by “then”
17) Page 6, line 249: “…potential biotechnological applications”, such as? It would be nice to see some examples
18) Pages 11 and 12: Figures 5a and 5b are barely readable. Please replace by a better resolution file.
Round 2
Reviewer 1 Report
The authors have addressed all of my concerns with the previous version.And hence, I support its publication in the current version.